# Parameters of Flow through Paravalvular Leak Channels from Computational Fluid Dynamics Simulations—Data from Real-Life Cases and Comparison with a Simplified Model

**DOI:** 10.3390/jcm11185355

**Published:** 2022-09-13

**Authors:** Michał Kozłowski, Krzysztof Wojtas, Wojciech Orciuch, Grzegorz Smolka, Wojciech Wojakowski, Łukasz Makowski

**Affiliations:** 1Department of Cardiology and Structural Heart Diseases, Medical University of Silesia, Ziołowa 47, 40-635 Katowice, Poland; 2Faculty of Chemical and Process Engineering, Warsaw University of Technology, Waryńskiego 1, 00-645 Warsaw, Poland

**Keywords:** paravalvular leak, computational fluid dynamics, hemolysis

## Abstract

Background: Shear forces affecting erythrocytes in PVL channels can be calculated with computational fluid dynamics (CFD). The presence of PVLs is always associated with some degree of hemolysis in a simplified model of the left ventricle (LV); however, data from real-life examples is lacking. Methods: Blood flow through PVL channels was assessed in two variants. Firstly, a PVL channel, extracted from cardiac computed tomography (CCT), was placed in a simplified model of the LV. Secondly, a real-life model of the LV was created based on CCT data from a subject with a PVL. The following variables were assessed: wall shear stress (*τ_w_*) shear stress in fluid (*τ*), volume of PVL channel with wall shear stress above 300 Pa (*V*_300_), duration of exposure of erythrocytes to shear stress above 300 Pa (*Vt*_300_) and compared with lactate dehydrogenase (LDH) activity levels. Results: *τ_w_* and *τ* were higher in the simplified model. *V*_300_ and *Vt*_300_ were almost identical in both models. Conclusions: Parameters that describe blood flow through PVL channels can be reliably assessed in a simplified model. LDH levels in subjects with PVLs may be related to *V*_300_ and *Vt*_300_. Length and location of PVL channels may contribute to a risk of hemolysis in mitral PVLs.

## 1. Introduction

Paravalvular leaks (PVLs) after surgical valve replacement represent an important complication. The presence of PVLs may lead to hemolysis and/or heart failure symptoms that require treatment [1]. The majority of PVLs (74%) develop during the first year after surgery [2]. 

Heart failure symptoms that may occur in the setting of a PVL are strictly related to the regurgitant volume. On the other hand, it seems that the pathogenesis of hemolysis is multifactorial [3]. This pathology is common in subjects qualified for percutaneous closure of PVLs—in one registry 50 out of 79 patients who underwent PVL closure had preprocedural hemolysis [4]. One of the factors that contributes to the development of hemolysis is shear stress, which is the component of stress coplanar with the material cross-section. It has been determined in previous studies that when shear stress exceeds 300 Pa, erythrocyte destruction can be observed [5]. Computational Fluid Dynamics (CFD) can today be utilized to determine the distribution of shear stresses in PVL channels. CFD is a numerical modelling method based on solving equations of differential balance of mass, momentum and energy for a small section of fluid, known as a computational cell. 

We have recently reported that CFD can be used to perform simulations of blood flow through mitral PVL channels [6]. We discovered that for relatively small PVL channels (with CSA of 3.6, 5.03, 7.9 and 20 mm^2^), the critical value of 300 Pa is always exceeded for the wall shear stress and shear stress in the fluid regardless of left-ventricular function. Our study had several limitations. The LV used for analysis was a model and was not extracted from a subject with real pathology. The shapes of PVL channels were created manually based on the available literature data. In addition, the presence of the left atrium walls was not accounted for. Therefore, we decided to create a model for flow analysis based on cardiac computed tomography data from a subject with a mitral PVL. In addition, we wanted to compare whether calculated parameters would differ between a real-life model and a simplified model that we used previously. Finally, we decided to give clinical context to our research by comparing calculated parameters that describe blood flow with lactate dehydrogenase activity (LDH) levels of subjects with PVLs.

## 2. Materials and Methods

To create PVL models for evaluation, a cardiac computer tomography (CCT) with contrast was used. To create the simplified model, an anonymous CT data of a healthy left ventricle that was already present in our database was extracted. To create a real-life model, retrospective CCT data from a subject with a mitral paravalvular leak was used. Semi-automatic segmentation of CCT data was performed in 3D Slicer software (https://www.slicer.org, accessed on 12 January 2020) [7].

Two 3D models of the left ventricle were created—CT-based (Figure 1, Table 1) and a simplified one (Figure 2, Table 1). The models consisted of the chamber of the left ventricle, an aortic outlet and a left atrium. Dimensions of the aorta and pulmonary veins of the simplified model were taken from [8] and [9], respectively. The volume of the left ventricle was 69 and 87 mL for the CT-based and simplified model, respectively. For the simplified model, the volume was determined based on average end-diastolic and end-systolic volume from CCT. Then, PVLs of various shapes, sizes and locations around the mitral annulus were introduced (Figure 3). Each PVL used in the study was extracted from CCT data of subjects with real mitral PVLs. The study consisted of two stages: (1) Firstly, one PVL shape (PVL I) was tested in two configurations of the left heart. (2) Secondly, based on results of the first stage, other PVL shapes were tested solely in the simplified model. Dimensions of all PVLs are described in Table 2. All simulations were performed in systole; therefore, the aortic valve was opened, the mitral valve was closed and backward flow towards the left atrium occurred only through the PVL channel.

Intraventricular blood flow was simulated using the commercial CFD software ANSYS Fluent 2020R1. The left ventricle’s numerical mesh used in simulations consisted of approximately 500,000 poly-hexcore cells, with an average size of 1 mm. In the zone surrounding the channel between the left ventricle (LV) and left atrium (LA), the mesh was of higher density, with an average cell size of 0.15 mm. The performed tests demonstrated that further mesh refinement did not affect the results—velocity profiles and energy dissipation rate in the system were identical (approx. 1–2% relative difference between meshes). Blood was assumed to be a non-Newtonian fluid with a density of ρ = 1060 kg/m^3^ and viscosity given by the Carreau–Yasuda model [10]:(1)$$\mu =0.0035+\left(0.1565\right){\left[1+{\left(8.2\gamma \right)}^{0.64}\right]}^{-1.23}$$
where γ is the shear rate and viscosity is given in Pa s.

Geometries of PVLs that were studied are shown in Figure 3. The length of the PVLs varied from 3.25 to 9.5 mm. Simulations were performed under transient-state conditions (pressure differences driving flow through the PVL channel were constant). For the purpose of this study, an 80 mmHg pressure gradient was chosen to perform all calculations, and a heart rate of 60 beats per minute was assumed (systolic phase lasted 0.3 s). Calculations were performed for the following blood flow rate profile [11]:(2)$$Q\left(t\right)=\{\begin{array}{c} 0.5Q_{S}\left(1-\cos\left(10\pi t\right)\right)\;\;\;for\;t\leq 0.1\\ 0.5Q_{S}\left(1+\cos\left(5\pi \left(t-0.1\right)\right)\right)\;for\;0.1\leq t\leq 0.3\\ \;\;\;\;\;\;\;\;\;\;\;\;\;\;\;\;\;\;\;\;\;\;0\;\;\;\;\;\;\;\;\;\;\;\;\;\;\;\;\;for\;t\geq 0.3 \end{array}$$
where $Q_{S}$ = 414 mL/s is the peak blood flow rate.

The following variables were examined regarding blood flow through the PVL channel: wall shear stress, shear stress in fluid, volume of PVL channel in which wall shear stress exceeded 300 Pa, and duration of exposure of red blood cells in the PVL channel to shear stress values above 300 Pa. In addition, lactate dehydrogenase levels (LDH) of subjects with mitral PVLs who provided CCT data for model preparation are also presented and correlated with obtained parameters that describe blood flow. This study did not require the approval of an Ethics Committee.

## 3. Results

Changes of the maximal wall shear stress during systole in both models for PVL I are presented in Figure 4. The values of maximal wall shear stress were higher for the simplified model during the whole systolic phase. 

Changes of maximal shear stress in fluid during systole in both models for PVL I are presented in Figure 5. Values of maximal shear stress in fluid were higher for the simplified model during the whole systolic phase.

Changes in volume of the PVL channel, in which the shear stress exceeded 300 Pa during systole in both models for PVL I, are presented in Figure 6. The obtained values were almost identical in both heart models.

Changes in the duration of exposure of red blood cells to shear stress above 300 Pa during systole in both models for PVL I are presented in Figure 7. The obtained values were almost identical in both heart models.

Changes of maximal wall shear stress during systole for all PVL models are presented in Figure 8. Highest values were obtained for model III during the whole systolic period.

Changes of maximal shear stress in fluid during systole for all PVL models are presented in Figure 9. The highest values were obtained for model III during the whole systolic period.

The changes in volume of the PVL channel, in which shear stress exceeded 300 Pa during systole for all PVL models, is presented in Figure 10. Almost identical values were obtained for models I, III and IV, whereas significantly lower values were observed in the case of model II.

Changes in the duration of exposure of red blood cells to shear stress above 300 Pa during systole for all PVL models are presented in Figure 11. The highest values were obtained for model III and the lowest for model II during the whole systolic period.

LDH levels for each subject are presented in Table 3. 

## 4. Discussion

In this study, we analyzed several parameters that describe the flow of blood through four PVL channels of various sizes. Initially, all calculations were performed for the PVL I model for both the real-life geometry and the simplified one. Based on the results obtained during this phase, we decided to perform all further calculations in the simplified model (explanation below). All cases represent real-life pathologies treated in our center.

The results of the flow assessment through the analyzed PVL I channel for both models demonstrate that values of maximal wall shear stress and maximal shear stress in the fluid are slightly but consistently higher during the systolic phase for the simplified model. These values, however, are just maximal measured values of shear stress that were obtained for a small number of computational cells. Almost identical values were obtained for volumes of PVL channel in which shear stress exceeded 300 Pa as well as the duration of exposure of red blood cells to shear stress above 300 Pa in both heart models. This indicates that a similar number of red blood cells should be destroyed in both cases. It is therefore safe to state that, for the purpose of blood flow assessment through a PVL channel, there is no difference in values of parameters that relate to hemolysis severity between the simplified and real-life LV model. This is an important observation, since the most time-consuming part of the CFD assessment is model preparation. Our results suggest that one universal model of the left ventricle can be used as a template and only individual PVL morphologies could be extracted from CCT or echocardiographic data and introduced to the simplified model. This approach will expedite research in this area, since a higher number of cases can be analyzed in the same amount of time. 

Based on these results, simulations of flow through PVL channels II, III and IV were performed solely in a simplified model. For this study, we chose cases with larger CSAs—the smallest PVL had CSA of 15.7 mm^2^ and the largest had CSA of 32 mm^2^. These PVL models were extracted from subjects that were referred to our center for the evaluation of detected PVLs. Contrary to our previous study, the maximal wall shear stress and maximal shear stress in fluid values were not related to CSA of a PVL channel. The highest values were reported for the model IV and lowest for model II. This may have different explanations. Firstly, the PVL channels used in this study differed in length, while previously, a fixed length of 3 mm was utilized. It is interesting to note that for all analyzed parameters (maximal wall shear stress, maximal shear stress in fluid, volume of the PVL channel in which shear stress exceeds 300 Pa and duration of exposure of red blood cells shear stress values above 300 Pa) the lowest values were observed for model II, which had the longest channel. This was unexpected, since the volume of model II was similar to model I and III. PVL channel length is not routinely measured in a clinical setting, because it is not useful in grading PVL severity. Until now, the only practical indication for PVL channel length measurements was optimal closure device selection, since some devices, such as the Occlutech PLD device (Occlutech Holding, Switzerland) are not suitable for PVL channels with a length greater than 3 mm. However, our current observations suggest that channel length may be related to the risk of hemolysis in a PVL setting.

Another possible factor that might explain differences in analyzed parameters between all PVL models is the location of PVL channels around the mitral annulus. As depicted in Figure 2, in models I, III and IV, the channel is located anteriorly, whereas in model II, a posterolateral location can be described. During systole, the velocity of blood in the left ventricular outflow tract is higher than in the chamber of the left ventricle. This higher velocity corresponds to higher shear stress. This is depicted in Figure 12, where the red box number 1 represents the location of anterior PVL channels and the red box number 2 represents the posterolateral location. When the isovolumetric phase of left-ventricular contraction ends, the aortic valve opens and blood flows through the left ventricular outflow tract to the aorta. If a PVL channel is present in the anterior location of the mitral annulus, backward flow to the left atrium occurs and due to higher blood velocity, a greater shear stress will be observed.

To support our experimental findings, a clinical context is required. The presence of hemolysis is usually diagnosed with a combination of laboratory tests. LDH activity, hemoglobin concentration and reticulocyte count are among the most commonly used parameters to search for hemolysis in a clinical setting. We observed the lowest LDH activity in the case of a PVL model II, which had the smallest calculated volume of the PVL channel, in which shear stress exceeded 300 Pa as well as the shortest duration of blood exposure to shear stress values above 300 Pa. On the other side of the spectrum were PVL models I, III and IV, which had the highest and similar calculated volumes of PVL channels, in which shear stress exceeded 300 Pa. It is, however, interesting to note that the duration of blood exposure to shear stress values above 300 Pa differed between these three models, and the longest duration was observed in the case of model III, which was created from the CCT data of a subject with the highest LDH levels. We speculate that at least two possible scenarios are possible to achieve clinically significant hemolysis. First, it is possible when the volume of a PVL channel in which shear stress exceeds 300 Pa is very high. In such case, even when the duration of the exposure of blood to such shear stress is short, a significant number of erythrocytes can be destroyed. Second, it is possible when the duration of exposure of red blood cells to shear stress above 300 Pa plays a dominant role, and in such cases, even small volumes with increased shear stress may be enough. Interplay between these parameters seems to be intricate and further studies with real-world examples are required to test all hypotheses. 

In our opinion, the simulations we performed have certain clinical applications. A significant number of patients leave surgical wards with paravalvular leaks, which are deemed to not be clinically significant at discharge. By identifying factors that predispose to hemolysis in paravalvular leaks (such as channel length or location), surgeons may be more willing to correct leaks that appear insignificant in periprocedural echocardiography, but carry risk of hemolysis in the long term follow-up. This thinking may also apply to follow-up strategies, where physicians may schedule visits more often for patients with PVLs known to more likely cause hemolysis. We also speculate that with increased heart rhythm and/or increased pressure gradient between cardiac chambers, higher shear stress should occur. In such cases, patients could benefit from changes in pharmacotherapy. We are currently preparing new models of cardiac chambers to test that hypothesis.

## 5. Conclusions

An analysis of parameters that describe blood flow through PVL channels yields comparable results between a simplified heart model and a real-life model. LDH levels in subjects with PVLs may be related to the volume of a PVL channel, in which the shear stress exceeds 300 Pa as well as the duration of blood exposure to shear stress values above 300 Pa. Both length and location of a PVL channel may contribute to a risk of hemolysis in mitral PVLs.

## Figures and Tables

**Figure 1 jcm-11-05355-f001:**
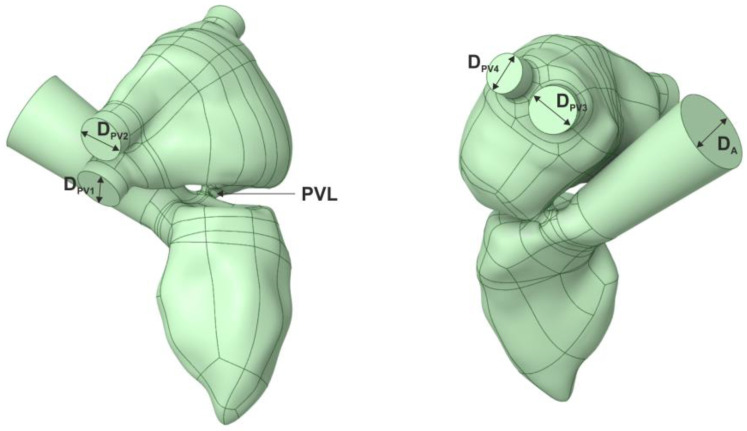
CT-based model of the left ventricle.

**Figure 2 jcm-11-05355-f002:**
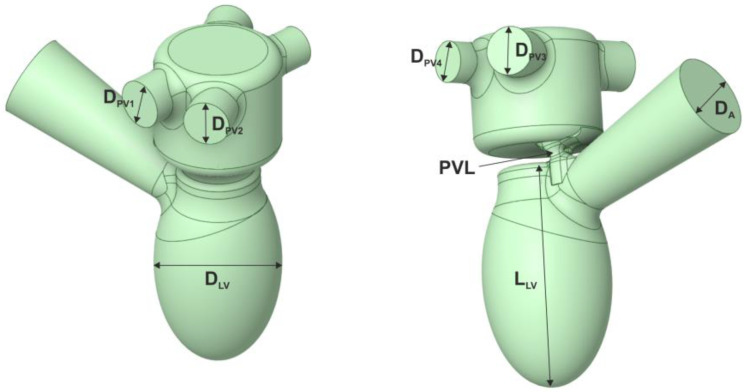
Simplified model of the left ventricle.

**Figure 3 jcm-11-05355-f003:**
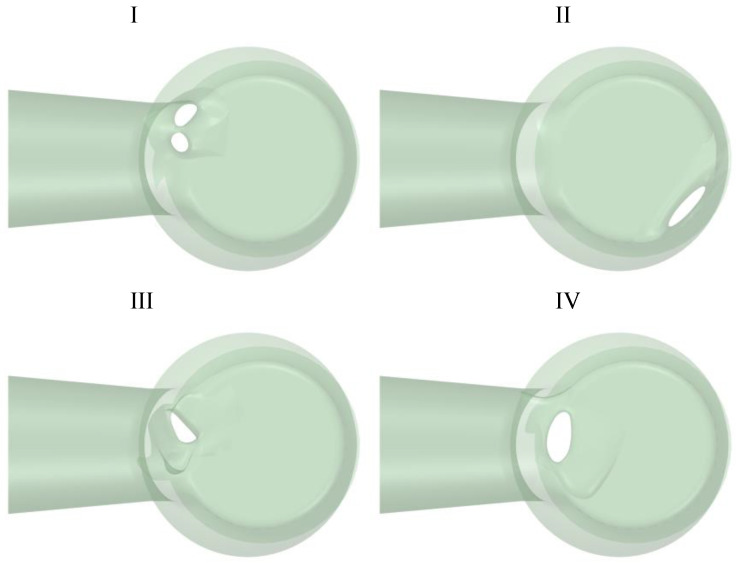
PVL geometries. (**I**,**III**,**IV**)—anterior mitral leaks, (**II**)—posterolateral mitral leak.

**Figure 4 jcm-11-05355-f004:**
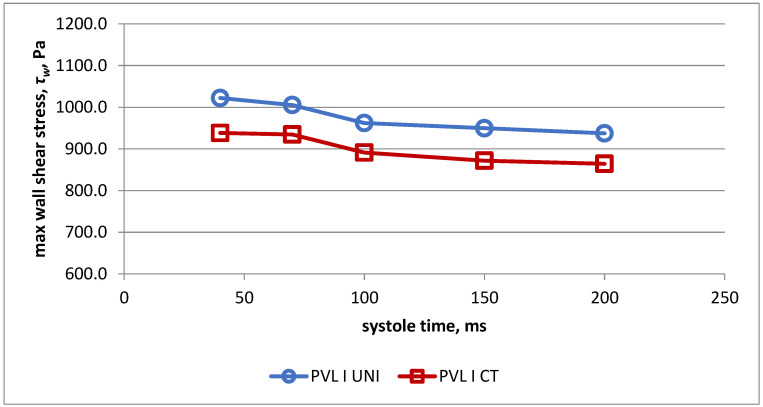
Values of maximal wall shear stress in universal model (UNI) and real-life model (CT) for PVL model I.

**Figure 5 jcm-11-05355-f005:**
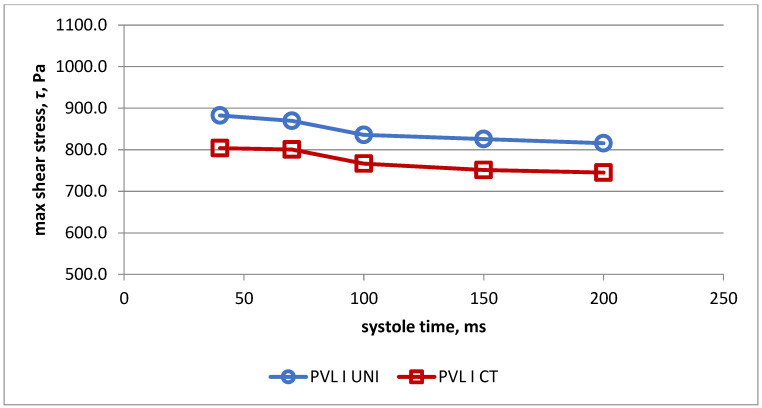
Values of maximal shear stress in fluid in universal model (UNI) and real-life model (CT) for PVL model I.

**Figure 6 jcm-11-05355-f006:**
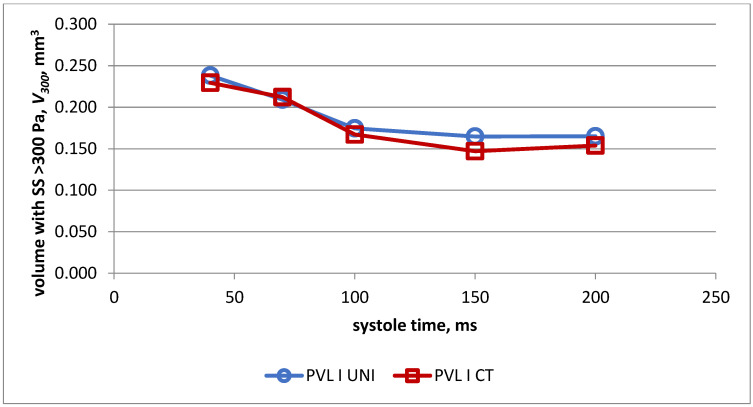
Volume with shear stress > 300 Pa in universal model (UNI) and real-life model (CT) for PVL model I.

**Figure 7 jcm-11-05355-f007:**
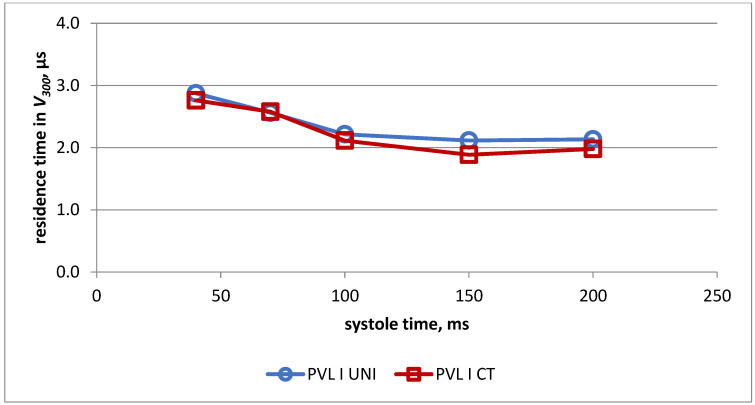
Duration of exposure to shear stress > 300 Pa in universal model (UNI) and real-life model (CT) for PVL model I.

**Figure 8 jcm-11-05355-f008:**
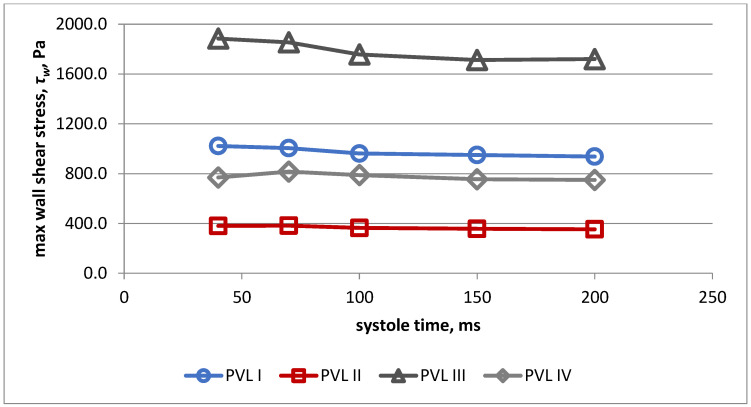
Maximal wall shear stress during systole for all PVL models.

**Figure 9 jcm-11-05355-f009:**
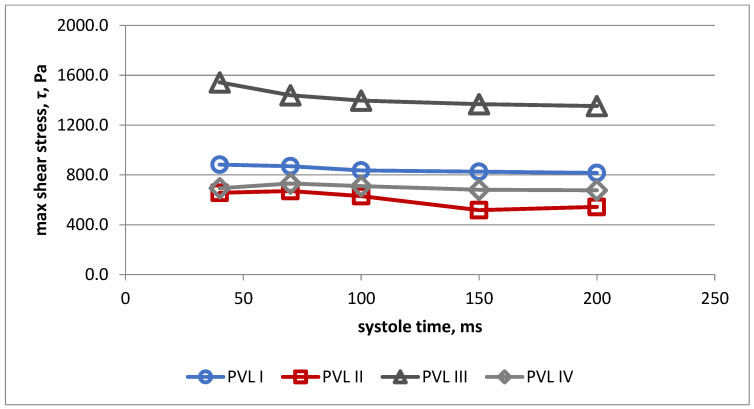
Maximal shear stress in fluid during systole for all PVL models.

**Figure 10 jcm-11-05355-f010:**
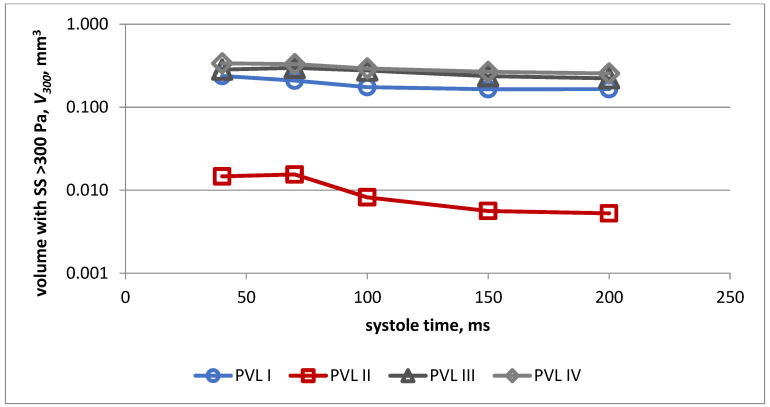
Volume of the PVL channel in which shear stress exceeded 300 Pa during systole for all PVL models.

**Figure 11 jcm-11-05355-f011:**
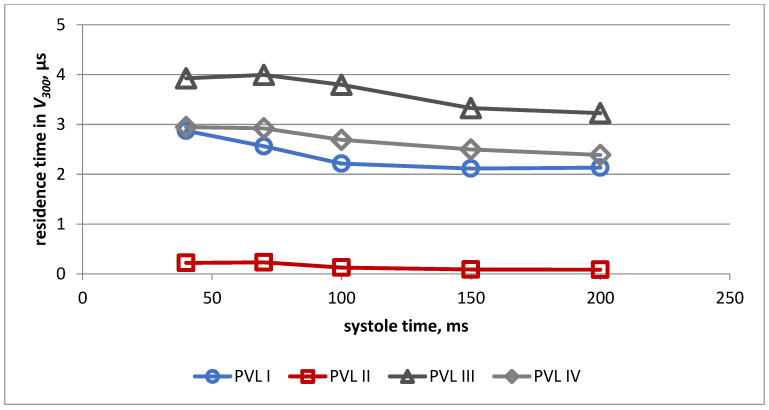
Duration of exposure to shear stress > 300 Pa during systole for all PVL models.

**Figure 12 jcm-11-05355-f012:**
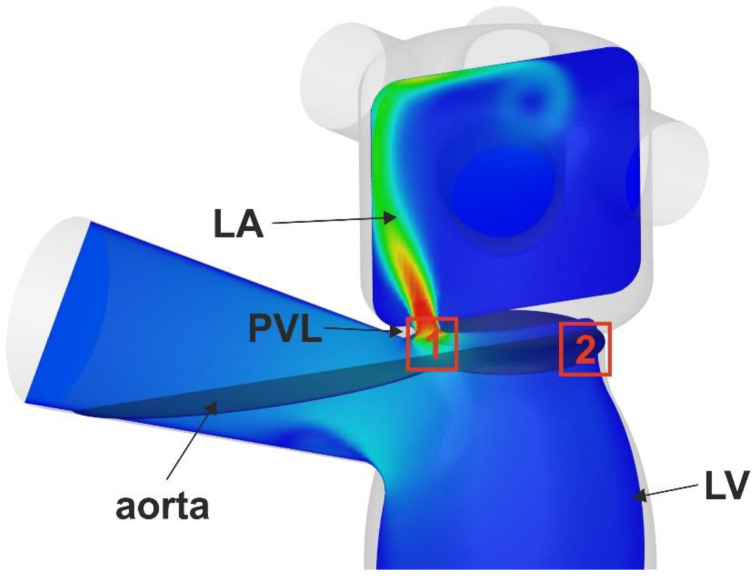
Velocity distribution in the model of the left ventricle, left atrium and PVL channel. Boxes 1 and 2 correspond to anterior (PVL I, III, IV) and posterolateral (PVL II) locations, respectively.

**Table 1 jcm-11-05355-t001:** Dimensions of the models of the left ventricle.

Dimension	Size
*D_A_*	30 mm
*D_PV_* _1_	17.4 mm
*D_PV_* _2_	16.5 mm
*D_PV_* _3_	16.95 mm
*D_PV_* _4_	14.45 mm
*D_LV_*	37.5 mm
*L_LV_*	65 mm

**Table 2 jcm-11-05355-t002:** Characteristics of PVL models.

Model Number	Total Cross-Sectional Area (mm^2^)	PVL Circumference (mm)	PVL Volume (mm^3^)	PVL Wall Area (mm^2^)	PVL Length (mm)
I	26.2	27.0	269.8	242.3	6.5
II	15.7	20.2	237.8	279.5	9.5
III	20.4	17.7	279.7	210.1	7.3
IV	32.0	22.6	93.8	97.5	3.25

**Table 3 jcm-11-05355-t003:** LDH levels in subjects with respective PVLs.

Model Number	LDH (U/l)
I	560
II	285
III	809
IV	441

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
