# Peer review of "Parameters of Flow through Paravalvular Leak Channels from Computational Fluid Dynamics Simulations—Data from Real-Life Cases and Comparison with a Simplified Model"

_jcm, 2022, doi:10.3390/jcm11185355_

Round 1
Reviewer 1 Report
This is a research study in which the authors tried to create a model for flow analysis with CFD based on cardiac computed tomography data from a subject with a mitral para-valvular leak (PVL). The authors also compared if the calculated parameters would differ between a real-life model and a simplified model. The 3rd goal was to compare calculated parameters that describe blood flow with lactate dehydrogenase activity (LDH) levels of subjects with PVLs. All calculations were done on two 3D models of the left ventricle with four PVL channels of various sizes.
The results showed that there was no difference in the results between a simplified heart model and a real-life model. LDH levels in subjects with PVLs may be related to volume of a PVL channel in which shear stress exceeds 300 Pa as well as the duration of blood exposure to shear stress values above 300 Pa. Both length and location of a PVL channel may contribute to risk of hemolysis in mitral PVLs.
Good methodology
Results relevant to current research and medical or surgical management of PVL.
Author Response
Thank you for your review. The reviewer had no comments regarding composition of our work therefore we don't attach any responses.
Reviewer 2 Report
In this interesting paper, Kozlowski and colleagues explore fluid dynamics in patients with paravalvular leaks following mitral valve surgery. The paper is well written and presents interesting data.
I only have some minor points:
A) how would sheer forces change, if the applied pressure gradient and/or heart rate vary? This analysis could increase the value of the paper by suggesting non-interventional ways to limit hemolysis (e.g. more aggressive rate control, volume management etc).
B) apart from mechanistic understanding of the forces behind hemolysis in PVL, does such a simulation yield an added benefit in the clinical management?
C) Figure 12 may be improved by adding labels to the different anatomic structures and specifying orientation (septal/lateral), as well as including legend for PVL locations 1/2. Furthermore: the stress depicted here is in the presence of leak in which position (1, 2, both)?
Author Response
Thank you for your review and for interesting ideas that could improve our work.
Below please find responses to your comments
Comment: how would sheer forces change, if the applied pressure gradient and/or heart rate vary? This analysis could increase the value of the paper by suggesting non-interventional ways to limit hemolysis (e.g. more aggressive rate control, volume management etc).
Response: This is a valid question. We speculate that with increased heart rate and increased pressure gradient the shear forces would also increase. Unfortunately it is not possible to perform adequate calculations that could support our theory within such short timeframe that we have to provide our response (our engineers say it would take several weeks). We are, however, currently preparing another heart model to further evaluate this issue.
Comment: apart from mechanistic understanding of the forces behind hemolysis in PVL, does such a simulation yield an added benefit in the clinical management?
Response: In our opinion simulations we performed have certain clinical applications. A significant amount of patients leave surgical wards with paravalvular leaks which are deemed to be not clinically significant at discharge. By identifying factors that predispose to hemolysis in paravalvular leaks (such as channel length or location) surgeons may be more willing to correct leaks that appear insignificant in periprocedural echocardiography but carry risk of hemolysis in long term follow-up. This thinking may also apply to follow-up strategies where physicians may schedule visits more often for patients with PVLs known to more likely cause hemolysis. Future simulations may also provide advice regarding the optimal pharmacological strategies in subjects with PVL and hemolysis (as suggested in your previous comment).
Comment: Figure 12 may be improved by adding labels to the different anatomic structures and specifying orientation (septal/lateral), as well as including legend for PVL locations 1/2. Furthermore: the stress depicted here is in the presence of leak in which position (1, 2, both)
Response: Labels were added, caption was changed. Figure was reformated to better concentrate on region of interest. Since the orientation of the figure is anteroposterior we feel that adding labels for septal/lateral would create some confusion (it can be done of course). The stress depicted in the figure is in presence of PVL in position 1.
Round 2
Reviewer 2 Report
I thank the authors for their responses. Regarding point 2 (possible utility of such a simulation model in the clinical management of patients following mitral valve surgery), I would like them to briefly include this interesting perspective in the discussion.
I don´t have any further comments.
Author Response
A paragraph has been added in the discussion as requested.